# Conservation and Management of Protected Areas in China and India: A Literature Review (1990–2021)

Wen Gao [1,2,†], Jiefan Huang [2,3,†], Quan Qiu [2,4], Anil Shrestha [2], Changyan Yuan [2,5], Subhash Anand [6], Guibin Wang [1,*] and Guangyu Wang [2]

1   Center for Sustainable Forestry in Southern China, Nanjing Forestry University, Nanjing 210037, China
2   Faculty of Forestry, University of British Columbia, 2424 Main Mall, Vancouver, BC V6T 1Z4, Canada
3   College of Forestry, Hebei Agricultural University, Baoding 071001, China
4   College of Forestry and Landscape Architecture, South China Agricultural University, Guangzhou 510642, China
5   School of Economics and Management, Beijing Forestry University, Beijing 100083, China
6   Department of Geography, School of Economics, University of Delhi, Delhi 110007, India
*   Correspondence: gbwang@njfu.edu.cn
†   These authors contributed equally to this work.

**Abstract:** Protected areas (PAs) are key to biodiversity conservation. As two highly populous and biodiverse countries, China and India are facing similar socioenvironmental pressures in the management of PAs. A comparative analysis of studies of PA policies in these two countries provides an objective assessment of policy concerns. This study involved a bibliometric analysis of studies of PA policies in China and India. Relevant publications were retrieved from the Web of Science and Scopus. The analysis was carried out using the Bibliometrix R Package, CiteSpace, and VOSviewer. The results indicate that PA policies studies in China are growing at an exponential rate, while Indian studies were cited significantly more often. "Environmental protection" was the main focus of the Chinese studies, with top keywords including "forest ecosystem" and "strategic approach". In India, research was mainly focused on "wildlife management", and the top keywords were "climate change" and "ecosystem service". Studies from both countries were concerned with natural resource conservation and endangered species. Studies in India began relatively earlier and were more developed. India focused on people-related themes, while China emphasized strategic approaches. China is improving its system of PA and should learn from India to consider the relationship between environmental protection and people.

**Keywords:** conservation policy; bibliometric analysis; visualization; protected area; China and India

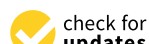



## 1. Introduction

Protected areas (PAs) play an essential role in preserving biodiversity and managing natural resources, [1]. They are also crucial for the culture and livelihood of indigenous and local communities [2]. National parks, sanctuaries, nature reserves, and other PAs play a prominent role in maintaining a healthy natural environment. PAs benefit millions of people by promoting tourism and offering protection against climate change and natural disasters by providing clean air and water [3]. The prominent ridge forest reserve, established by an ordinance of 13 April 1776, is the oldest PA in history, dating back 200 years [4]. The area of land protected by law has increased exponentially over the past 25 years [5]. More than 100,000 PAs have been established, accounting for 10.1–15.5% of the Earth's land surface (depending on the definition) [6,7].

As two emerging economic and populous giants, China and India are expected to be the most important countries in the world, which will largely determine the future environmental outcomes [8]. Over 45,000 plant species and 91,000 animal species, approximately 7–8% of the world's recorded species, live in India, a megadiverse country with

only 2.4% of the world's land area [9]. India's neighbor, China, also has extraordinary biodiversity, harboring more than 35,000 species of higher plants, 2700 terrestrial vertebrates and 28,000 marine species [10], with ecosystems ranging from permanent ice sheets to tropical humid forests [11]. In terms of biodiversity, they are both among the most biodiverse countries in the world [12,13]. The establishment of PAs is the primary strategy to preserve biodiversity [14]. The development of PAs is strongly related to the implementation of the policies [15,16]. The Plan of Natural Forest Nature Reserve Construction, China's first law on PAs, was enacted in 1956. Meanwhile, the first nature reserve in China, Dinghushan National Nature Reserve, was also established in the same year. Therefore, it is widely recognized that 1956 marked the beginning of modern nature conservation in China [17]. The first five national parks in China were announced in October 2021 [18]. Under the Principle for the Categories and Grades of Nature Reserves (GB/T 14529-93), China's PAs can be mainly divided into three major categories and nine types [19]. The first national park in India, presently named Jim Corbett National Park, was established in 1936 [20]. The Indian Forest Act (1927, IFA) and Wildlife Protect Act (1972, WA) provide legal guarantees for the establishment of India's nature reserve system [21]. Currently, official PAs in India include 104 National Parks and 551 Wildlife Sanctuaries [20]. Notably, China and India both established PAs in the Himalayan biodiversity hotspot and both play an important role in maintaining biodiversity in the Himalayan region [8].

As two populous countries in Asia, they face similar socioenvironmental pressures and have developed a range of policies to manage them. Hence, the PA policies of China and India deserve close scrutiny. Furthermore, a comparative analysis of China's protected area policy (CPAP) and India's protected area policy (IPAP) can identify the strengths and weaknesses in PA development of the two countries. Due to the imperfection of policy formulation or implementation, a large number of studies analyzed the respective PA policies of the two countries. A further review and summary of these publications can provide a fresh perspective on the regulation of PA policies in both countries. This research aims at looking back at the history of China and India's protected area, deepening the understanding of the original causes of their current development status. Our findings can be applied to the protected area policymaking of similar countries and make contributions to global biodiversity conservation.

This study uses bibliometric analysis to analyze publications related to PA policy in both countries on a qualitative and quantitative basis [22–24]. Accordingly, PA policies in India and China were systematically evaluated, analyzed, and compared over the past 32 years, highlighting the research directions for the future.

## 2. Materials and Methods

### 2.1. Search Strategy

Web of Science (WOS) was previously the only citation database that covered all areas of science for many years. However, WOS was challenged by Scopus since it was released in 2004, which is built on a similar breadth and scale [25]. Traditionally, Web of Science and Scopus are two of the most widely used databases for bibliometric analysis [26]. We, therefore, retrieved published data involving the CPAP and IPAP from the Web of Science Core Collection (WoSCC) and Scopus. The search time was set from 1 January 1990 to 31 December 2021 to include recent studies. We specially added search terms "tiger reserve*" and "panda reserve*", as they are essential components of PAs in China and India, respectively. Research article was the only document type included. The data search was conducted by combining the search string with Boolean operators (Table 1).

**Table 1.** Search strategies for India and China.

| Country | Database | Type of Dataset | Time Period/Timespan | Search Terms | Total No. of Publications |
|---|---|---|---|---|---|
| India | WoSCC and Scopus | Published and research articles | 1 January 1990–31 December 2021 | National park/protected area/nature reserve/tiger reserve/wildlife sanctuary AND policy/approach/strategy/program/guideline | 1301 |
| China | WoSCC and Scopus | Published and research articles | 1 January 1990–31 December 2021 | National park/protected area/nature reserve/panda reserve/wildlife sanctuary AND policy/approach/strategy/program/guideline | 1263 |

### 2.2. Data Collection

Publications from WoSCC were exported in plain text format (TXT), while publications from Scopus were exported in research information systems (RIS) format. We converted the RIS format to TXT in WoS using CiteSpace (version 5.8.R3). By comparing the two databases, duplicate papers were manually removed. This reduced the final number of publications from India and China used in the analysis to 1301 and 1263, respectively. All the information obtained from the two databases, including the number of papers and citations, titles, authors, affiliations, countries, keywords, journal, and publication year were collected and integrated for bibliometric analysis and visualization.

### 2.3. Statistics Analysis

The scientific analysis in this study focuses on three components for a comprehensive analysis of the hotspots, trends, and evolution of publications from 1990 to 2021 to compare the CPAP and IPAP. Three visualization tools CiteSpace (5.8.R3) [27], VOSviewer (1.6.18) [28], and Bibliometrix 3.2.1 R Package (https://www.bibliometrix.org, accessed on 15 March 2022.) [29] were used to explore the structure and development of the research field and visualize the results obtained. Researchers can quickly evaluate the PA policies of the two countries, including research status, research hotspots, and emerging trends [30].

The first component is overall distribution, which reveals the primary characteristics of the publications over 32 years and was investigated using annual trends. The Bibliometrix R Package was used to calculate the annual number of publications. We presented a holistic assessment of the annual publications, delineating the fundamental features with indicators including the total number of publications (TP), the total number of citations (TC), the average number of citations per publication (AC), and the H-index [31]. The publications demonstrate the contributions of authors and institutes. The total citations and average citations per publication reflect the quality and impact of scholars. The H-index is often correlated with the quality and quantity of academic output at the same time. A dual-map overlay of publication was also implemented. The journal dual map facilitates the analysis of publication portfolios that can further reveal journal types or critical points.

The second component is cooperation analysis, demonstrating the structure and dynamic relationships between countries/regions and institutes. This component was evaluated as a function influential countries/regions, wherein we identified the most influential countries/regions of relevant publications. We presented their cooperation network using the Bibliometrix R Package. Country cooperation is one of the most significant forms of cooperation. Furthermore, articles related to country cooperation normally perform well in numbers and citations [32]. A co-authorship network map was also established. VOSviewer was used to visualize complex co-authorship networks among institutes to reveal the leading institutes of the area and their dynamic cooperation network over time.

The third component is keyword analysis (including co-occurrence network, classification analysis, timeline views, keywords citation bursts, and thematic evolution), which provides a spatial description of how various keywords are related to each other [33,34]. We deleted "India" and "China" from keywords to intuitively analyze the relationships among different keywords. The co-occurrence and co-authorship networks were presented by VOSviewer. CiteSpace was used for clustering, building the timeline view, and analyzing

the keyword citation burst in order to visualize the domain knowledge and its trends [35]. We also analyzed the thematic evolution to elucidate the dynamic changes in CPAP and IPAP over various periods and to demonstrate the evolutionary trends.

## 3. Results

### 3.1. Overall Distribution

PA-related publications showed a general upward trend for IPAP and CPAP (Figure 1). We divided the whole period into three stages as a function of annual production and growth rates: 1991–2005 (initial stage), 2005–2015 (growing stage), and 2016–2021 (the exponential growth stage for CPAP). In the initial stage, fewer than 30 publications each year involved both IPAP and CPAP. IPAP studies were slightly more prevalent than CPAP studies in the early two stages. In the growing stage, the number of publications involving both CPAP and IPAP increased, showing an overall stable upward trend. During the third stage, the number of publications associated with CPAP increased significantly and surpassed those associated with IPAP.

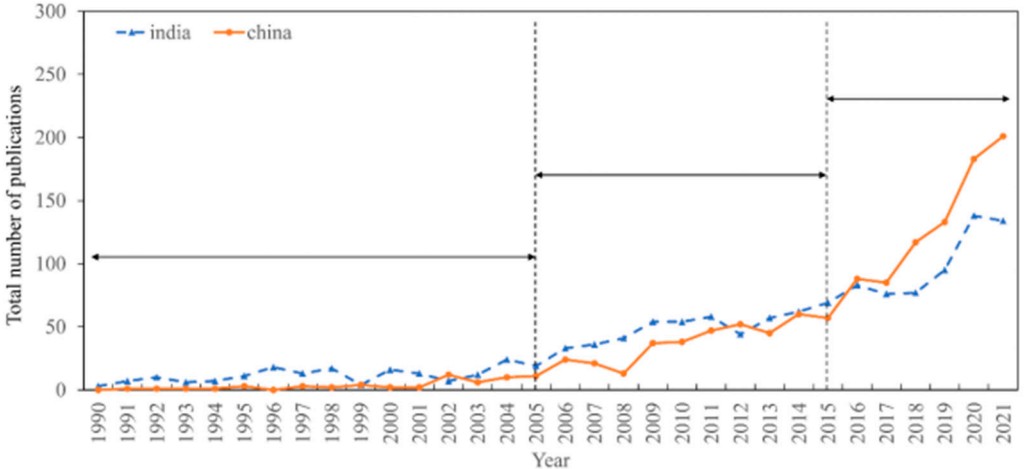

**Figure 1.** Annual trend in publications from 1990 to 2021.

Supplementary Table S1 presents the annual number of CPAP publications and citations from 1990 to 2021. The highest number of publications was 201 in 2021, with the highest TC of 800 in 2016, and the greatest AC of 95 in 2003. Furthermore, the highest percentage of TP (%TP) was observed in 2021. The maximum H-index of 14 was observed in 2018. Annual publications associated with IPAP were recorded from 1990 to 2021 (Supplementary Table S2). The highest TP of 138 was in 2020, and the highest TC of 1780 was in 2009. Furthermore, the greatest AC of 57.14 was found in 2002. In addition, 2020 showed the highest percentage of TP (%TP), while the maximum H-index of 23 was recorded both in 2009 and 2010. Overall, studies contributed significantly to the CPAP field in 2003, 2007, 2008, 2016, 2018, and 2021 and to the IPAP field in 2002, 2006, 2009, 2010, and 2020.

Analysis of the annual publication trends in CPAP and IPAP revealed a well-developed PA policy involving both CPAP and IPAP, with an increasing number of annual publications. CPAP was reported more often than IPAP in recent years; however, IPAP studies were cited significantly more often than CPAP, suggesting that IPAP studies have a higher academic value and significantly greater academic impact. Thus, the research analysis of IPAP was relatively more developed than CPAP.

The base map on the left reveals discipline clusters generated by mapping the citing journals, whereas discipline clusters of the cited journals are on the right. Research publications are, therefore, listed on the left and the references are listed on the right. Two key research citation paths were detected for CPAP (Figure 2a), while only one was associated with IPAP (Figure 2b). Journal publications involving CPAP and IPAP mainly focused

on "ecology, earth, and marine" fields. In contrast, most cited articles were published in journals related to "plant, ecology, and zoology" fields. The majority of the cited articles related to CPAP were also published in journals related to "economic and political" domains. Despite the strong association with the other two fields, these publications seem to be mainly inward-looking and isolated from outside fields. In order to break this trend, many challenges need to be overcome in this limited research ecosystem.

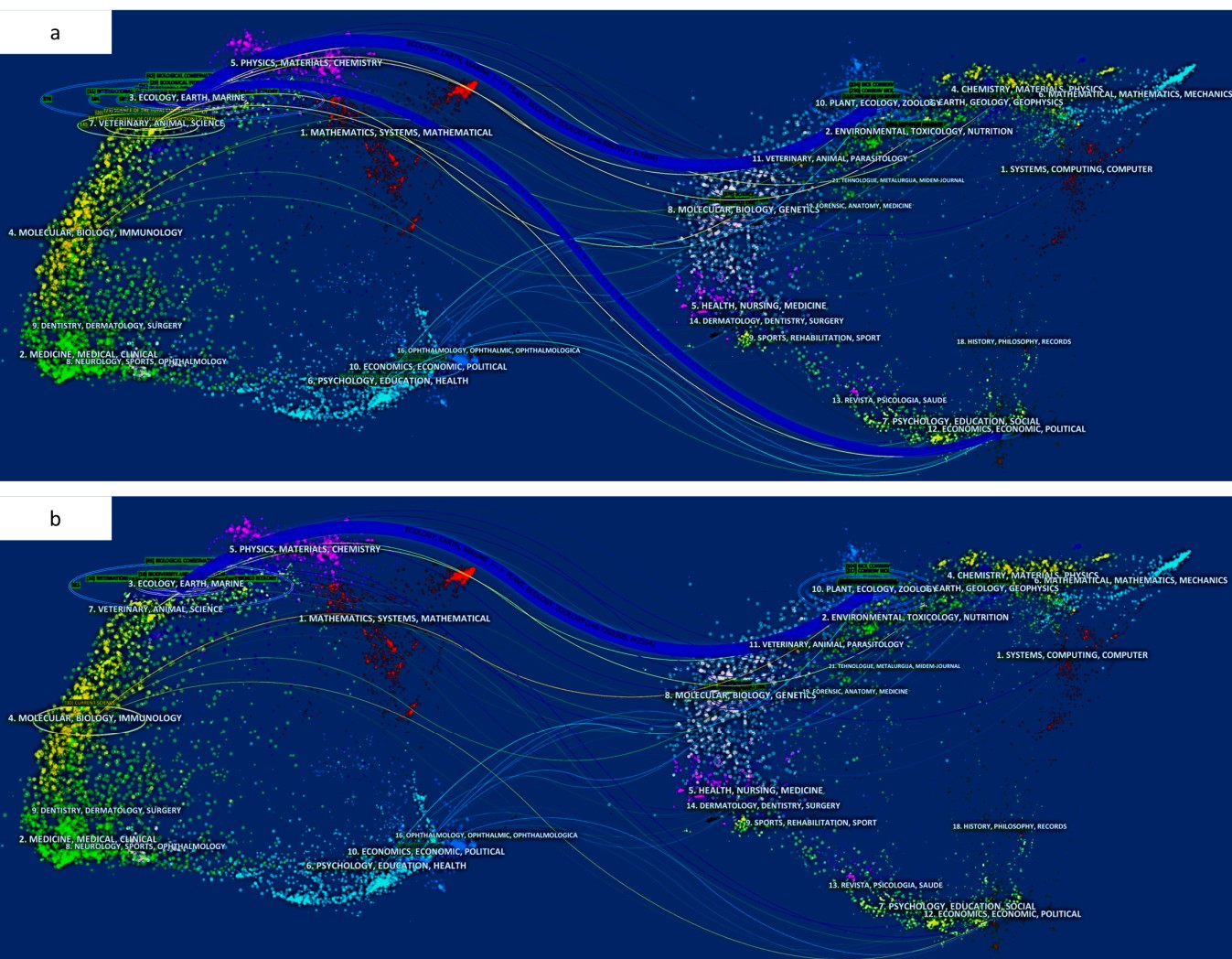

**Figure 2.** Dual-map overlay of articles citing China's protected area policy (CPAP) (**a**) and India's protected area policy (IPAP) (**b**) research. The left side indicates citing journals, the right side indicates cited journals, and the linkage paths denote the citation relationship.

### 3.2. Collaboration

A total of 55 countries/regions with university/research organizations participated in the publication of CPAP-related research articles (Figure 3a). They included eight countries/regions (except China) with more than 10 publications. This implies that these countries/regions emphasize research related to CPAP. In addition to China (1169 publications), the top three countries with the most publications are the USA (173 publications), the UK (34 publications), and Australia (33 publications). Overall, most of the studies related to PA were carried out by university and research organizations based in China with strong collaboration with countries worldwide. Furthermore, the USA (40 publications) had the most important academic ties with China, followed by Australia (nine publications) and the UK (nine publications).

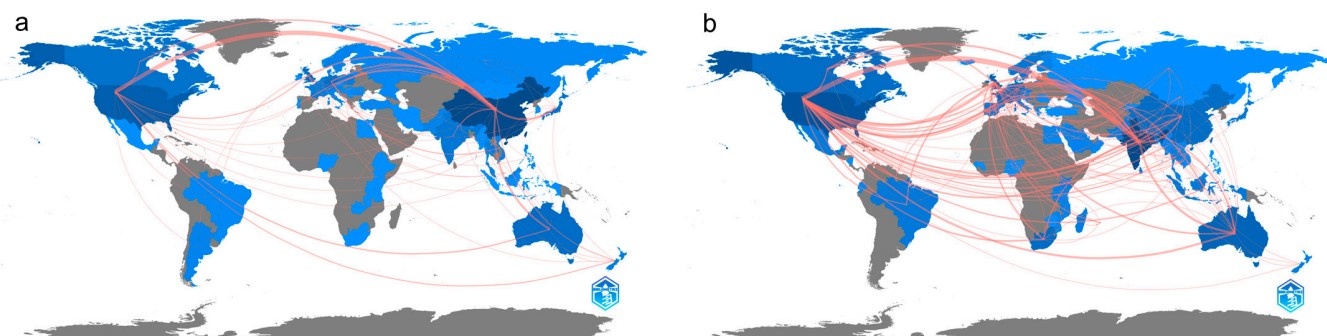

**Figure 3.** Geographical distribution of country/region collaboration in CPAP (**a**) and IPAP (**b**). The shaded colors indicate the volume of publications generated in the respective countries. The connections between countries reflect the cooperative relationship; thicker lines indicate further collaboration between countries.

Studies of IPAP have been conducted in 66 countries/regions (Figure 3b), which include 15 countries/regions (except India) with more than 10 publications. Additional countries are analyzing IPAP compared with CPAP, as reflected by the number of links in Figure 3. In addition to India (795 publications), the top three countries with the most significant number of publications were the same as those involved with CPAP, namely, the USA (293 publications), the UK (108 publications), and Australia (653 publications). Publications from the USA, the UK, and Australia involving IPAP outnumbered those analyzing CPAP. Overall, for IPAP, the level of collaboration between countries/regions was relatively higher than that of CPAP. Interestingly, the USA (29 publications) still had the strongest academic ties with India, followed by Australia (6) and the UK (5).

The visualization of collaboration networks reveals a relatively close cooperation between institutes, with slightly higher cooperation in CPAP than in IPAP. For CPAP, the University of Chinese Academy of Sciences is the leading institute with the highest number of collaborative relationships (Figure 4a). It has cooperated with almost all the other institutes. The Wildlife Institute of India is the leading institution in IPAP (Figure 4b). The cooperation between academic institutions in the field of CPAP increased after 2018, while that of IPAP decreased after 2016.

### 3.3. Keyword Analysis

The keyword co-occurrence network provides insight into the current hotspots and structural networks between keywords in CPAP and IPAP. As we can see in Figure 5, the selected keywords of CPAP and IPAP can be divided into five and four categories, respectively. Comparing the keyword co-occurrence networks of CPAP and IPAP reveals shared interest in conservation, namely, "protected area" and "national park". They also concentrate on keywords including "Asia", "Eurasia", "endangered species", "animalia", "ecosystems", and "human activities". Unlike IPAP, CPAP is strongly associated with biodiversity, which shows a very high word frequency in the co-occurrence map. CPAP also focuses on the "conservation of natural resources" and "Ailuropoda melanoleuca". By contrast, IPAP emphasizes marine conservation, with important keywords including "Indian ocean", "marine park", and "marine protected area". Moreover, IPAP is concerned with "wildlife", "tiger", and "humans".

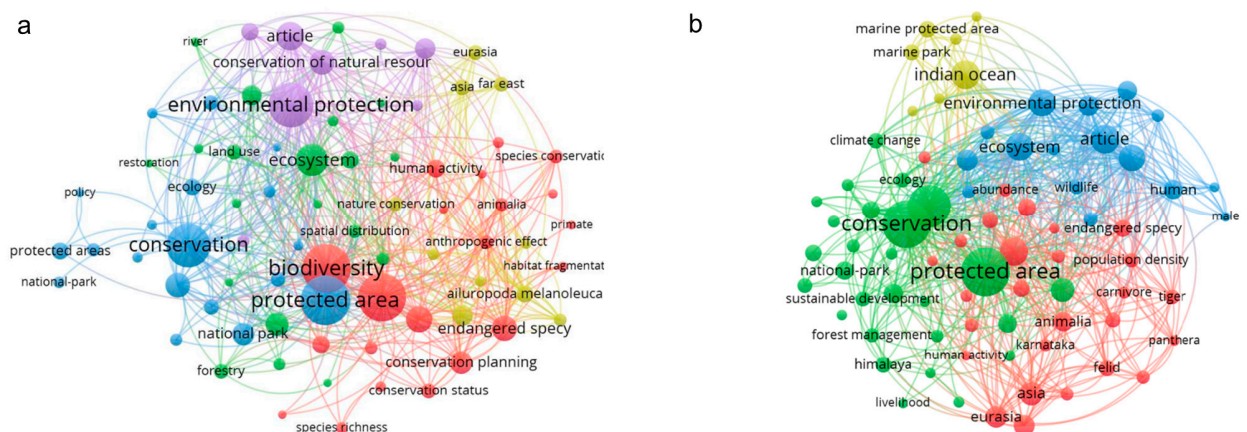

**Figure 4.** Timeline visualization of institute co-authorship networks for CPAP (**a**) and IPAP (**b**).

**Figure 5.** Keyword co-occurrence network of CPAP studies (**a**) and IPAP studies (**b**). The node size represents the frequency of occurrence, and each color represents a category.

The keyword timeline is presented to elucidate the hot topic trends in CPAP and IPAP at different periods (Figures Figure 6a and Figure 6b, respectively). For CPAP, 10 co-citation clusters are presented with their keywords. Every cluster is labeled in a specific period to represent the time distribution and precedence of the clusters. Cluster #0 "willingness to pay", cluster #2 "remote services", and cluster #4 "Ailuropoda melanoleuca" had the longest time span, lasting almost 30 years. They are still vital hotspots, with deepening study levels. Cluster #8 "conservation of natural reserves" was once a popular research topic (1991 to 2019). However, during the recent 2 years, no new articles were published on this theme. The time span of the clusters varied significantly for IPAP. Cluster #0 "remote sensing", cluster #1 "Indian ocean", cluster #2 "field", cluster #4 "human", and cluster #5 "nature conservation" had the longest time span, lasting almost 30 years. Cluster #0 "remote sensing", cluster #1 "Indian ocean", cluster #4 "human", and cluster #5 "nature conservation" are still active topics with a steady development.

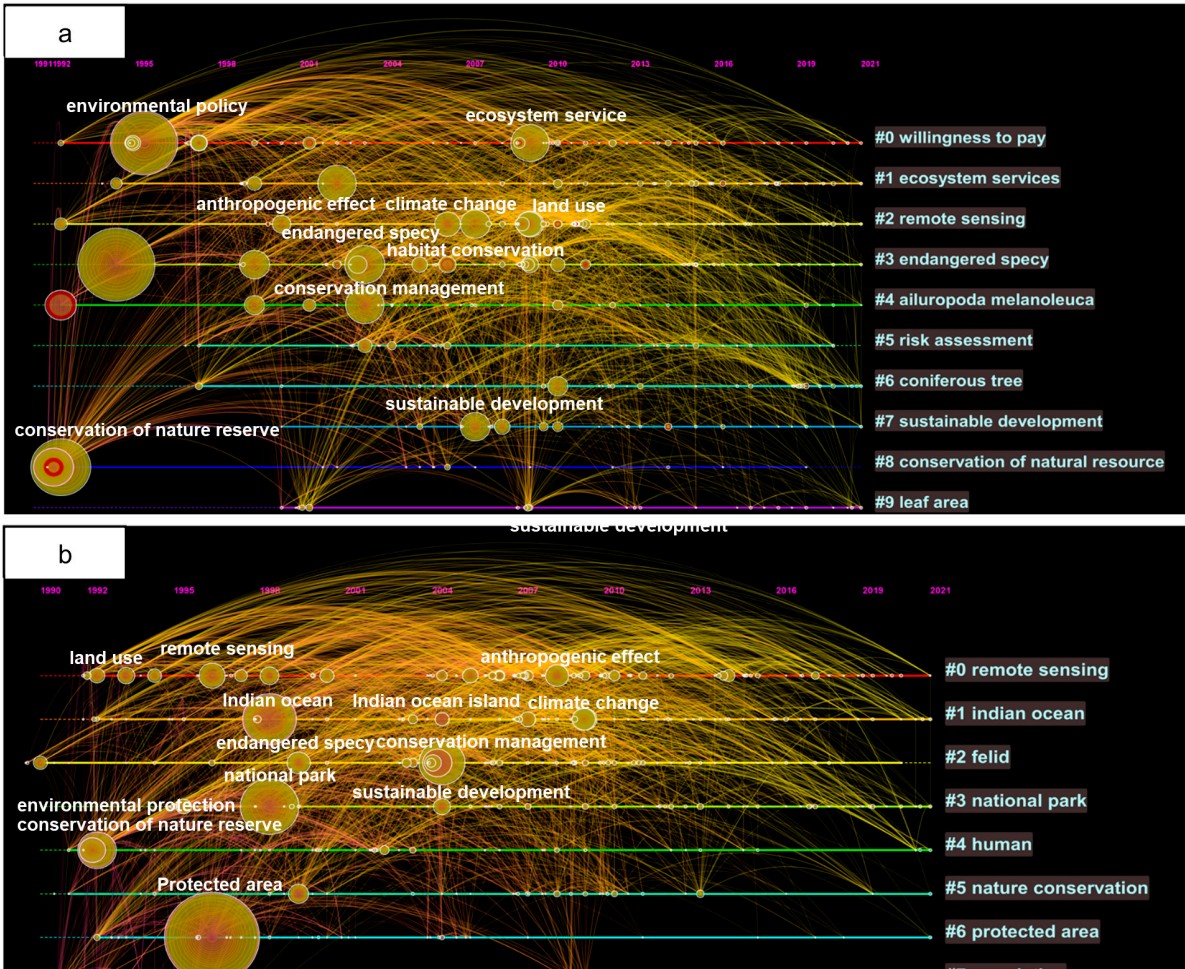

**Figure 6.** A timeline view of keywords in CPAP (**a**) and IPAP (**b**) research. The node locations demonstrate the first years of the keyword appearance. The node sizes represent the keyword frequency, and the connections between nodes represent the co-occurrence intensity. Important nodes (centrality > 0.1) are externally covered with the purple outer ring.

The map showing the timeline view, together with the curve linking these important nodes, indicates the central knowledge structure. CPAP focused more on "environmental policy", "ecosystem service", "endangered species", "habitat conservation", "conservation management", "sustainable development", and "conservation of nature reserve" (Figure 6a). Keywords changed continuously and became more diverse over time. Studies have increasingly emphasized the role of ecosystem, environment, and biodiversity conser-

vation in CPAP. In terms of India, IPAP focused more on the keywords "national park", "Indian ocean", "environmental protection", "conservation management", "remote sensing", "Indian ocean island", and "climate change" (Figure 6b). Studies investigating CPAP and IPAP used keywords such as "anthropogenic effect", "climate change", "land use", "endangered species", "conservation of nature reserve", and "sustainable development". This indicates that the two fields emphasized developmental hotspots in the corresponding field at different periods. We can see that CPAP focused strongly on environmental protection and decision making, while IPAP studies emphasized wildlife management and population, as well as ocean protection.

Citation burst detection demonstrates dynamic changes in keywords in the field over time. It can be used to determine the explosive data of academic interest (Pu and Qiu, 2015). In particular, the number of citations of some keywords increased sharply in a short period, which enables identification of fascinating points. Figure 7a,b list the information involving the top 10 keywords with the strongest citation bursts in research involving CPAP and IPAP.

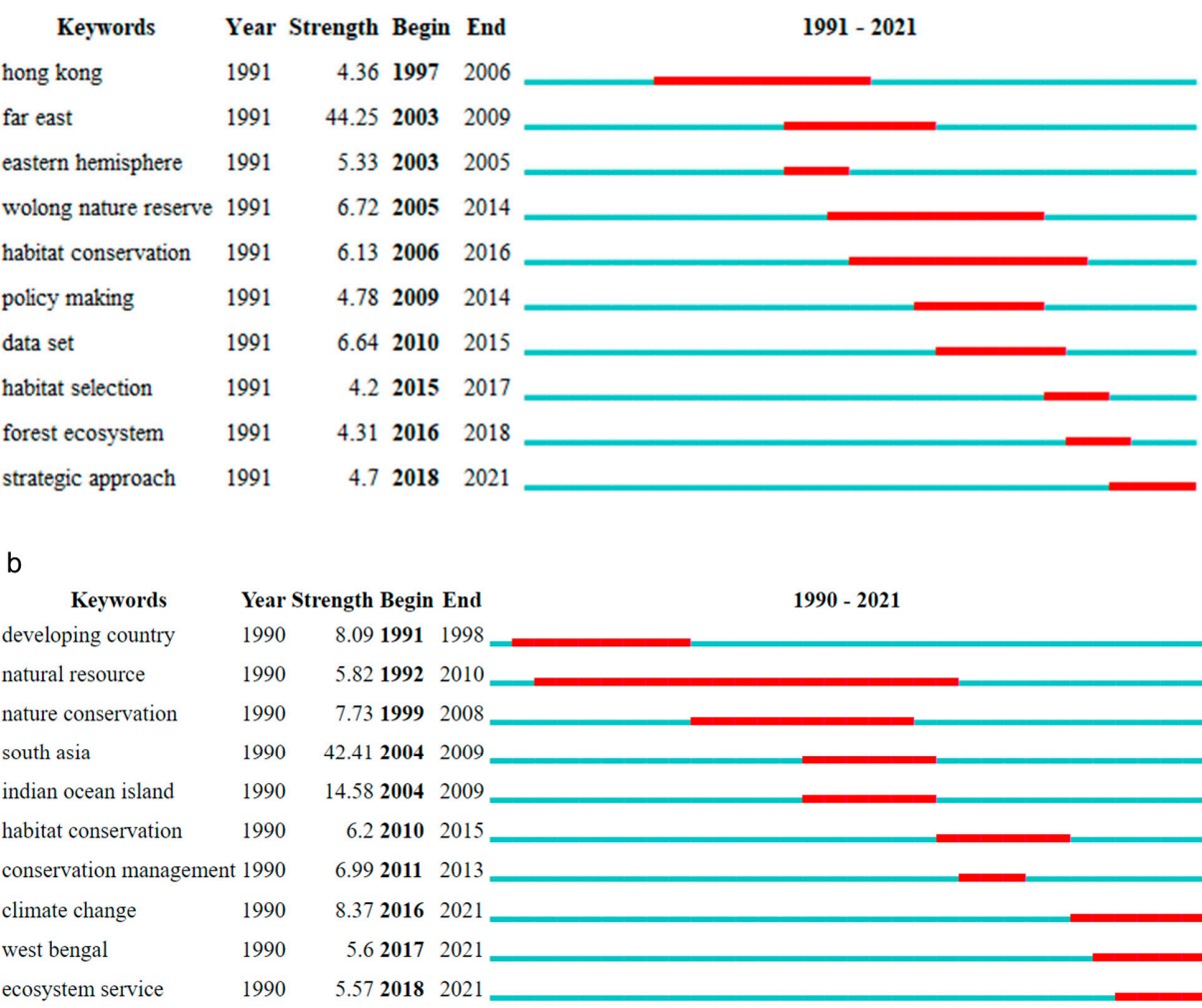

**Figure 7.** Top 10 keywords with the strongest citation bursts in CPAP (**a**) and IPAP (**b**) from 1990 to 2021.

The keywords cited with high intensity in CPAP could be divided into 10 categories, namely, regions ("Hong Kong", "Far East", and "Eastern Hemisphere"), PAs ("Wolong Nature Reserve", "habitat conservation", and "habitat selection"), the policy-related category ("policy development", "data setting", and "strategic approach"), and the natural resources-related category ("forest ecosystem"). The keywords mentioned in IPAP could

also be divided into 10 categories: regions ("South Asia", "West Bengal", "Indian Ocean Islands", and "developing countries"), the PA-related category ("natural resources", "nature conservation", and "habitat conservation"), the policy-related category ("conservation management"), and hot topics ("climate change" and "ecosystem services").

The category comparison shows that CPAP and IPAP focused on similar areas. However, CPAP continued to emphasize strategic approaches in the last 3 years, while IPAP focused on climate change, West Bengal, and ecosystem services in the previous 3 years. IPAP exhibited strong citation bursts for the keywords "natural resources" in 1992 and "nature conservation" in 1999, indicating that India began to focus heavily on natural resources much earlier than China. "Habitat conservation" in CPAP generated a strong citation burst in 2006, while IPAP started in 2010, indicating a large number of CPAP studies on habitat conservation very early compared with IPAP.

We divided the research period into four subperiods as a function of the publication volume to demonstrate the intellectual structure over time. Four relatively coherent strategy diagrams were generated using high-frequency keywords. The strategy map separates the various subjects into four distinct quadrants based on both centrality and density dimensions, providing us with a broad and intuitive idea of the research hotspots. The motor themes are critical for structuring policies of PAs and reveal a well-developed structure. The isolated themes indicate that the subject is highly specialized and is studied by fewer people. The emerging or declining themes are relatively weak compared with themes in other quadrants. Lastly, the fundamental themes are important for future research directions in the field of PA policy but have yet to receive adequate attention at this stage. The horizontal axis represents the degree of development, while the vertical axis is the degree of relevance.

The four consecutive subperiods were 1991–2002, 2003–2012, 2013–2018, and 2019–2021 (Figure 8). In the first subperiod (1991–2002), research related to "environmental protection" and "conservation" was well developed, and studies related to "protected area" were included in the basic theme. In the second subperiod (2003–2012), research on "article" was well developed, similar to "nature reserve", with emerging studies on "climate change". In the third subperiod (2013–2018), "environmental protection" research was well developed, "endangered species" studies were emerging, and "protected areas" studies were decreasing in popularity. In the fourth subperiod (2019–2021), research related to "environmental protection" was still thriving, while that on "biodiversity" and "forestry" represented key themes and direction for future research in the field. In addition, "human activity" was also investigated, yet these studies were relatively isolated and not well developed.

The first subperiod of research related to "protected area" was the dominant, fundamental, and transversal theme. In contrast, in the last two subperiods, "protected area" research gradually declined. The studies of "nature reserve", "endangered species", and "biodiversity" emerged. Note that "environmental protection", which was well developed in the first subperiod, continued to flourish in the last two subperiods, indicating the crucial role of CPAP-related investigations.

According to the strategic diagrams of IPAP-related publications (Figure 9), in the first subperiod (1990–2002), "protected area" was the basic theme, while "article" and "developing country" were well developed. In the second subperiod (2003–2012), the development of "protected area" research declined, and "environmental protection" studies were the drivers and well developed. At the same time, "biodiversity" research emerged and developed well. In the third subperiod (2013–2018), studies on "protected area" became the basic theme again, while research related to "conservation" and "tiger" emerged and developed well. In the fourth subperiod (2019–2021), "animal" research emerged as a major theme in the field and was well developed, while "protected area" research continued to develop and remain a primary theme for IPAP.

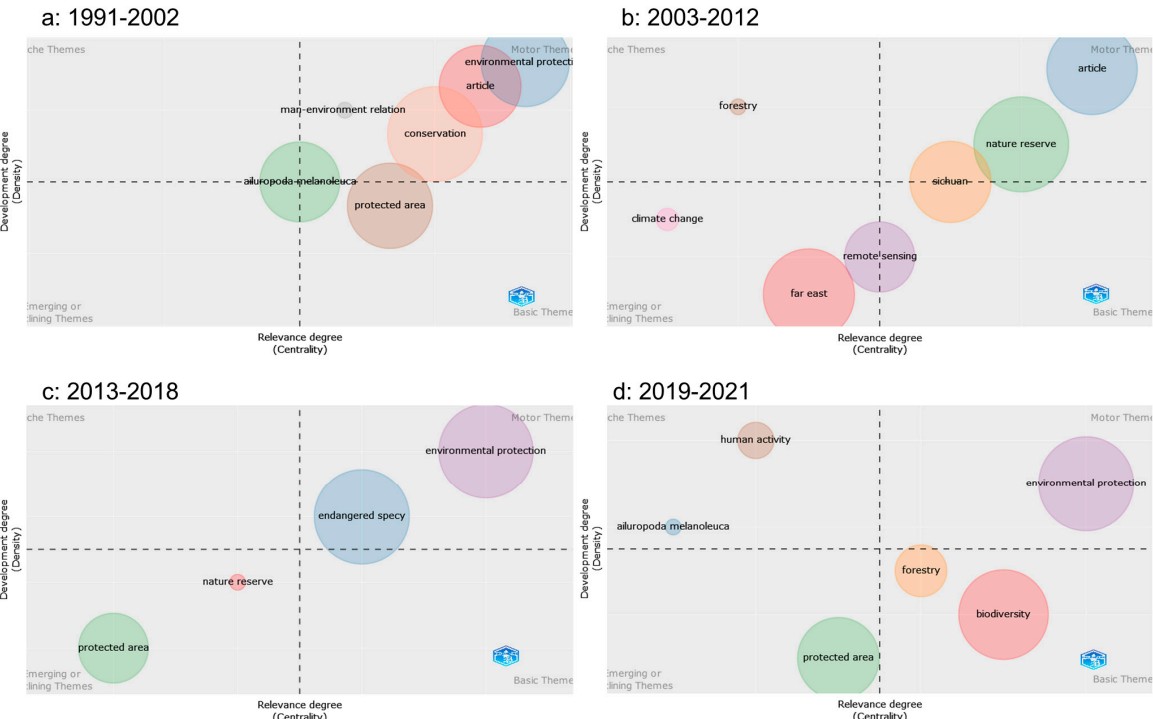

**Figure 8.** Strategic diagrams of the CPAP-related publications in 1991–2002 (**a**), 2003–2012 (**b**), 2013–2018 (**c**), and 2019–2021 (**d**). The upper-right quadrant represents motor themes; the upper-left quadrant indicates isolated themes; the lower-left quadrant showcases emerging or declining themes; the lower right quadrant involves fundamental themes. Each circle represents a topic, and the words on the circle are the keywords with the highest frequency involving the topic. The keyword frequency affects the size of the circle.

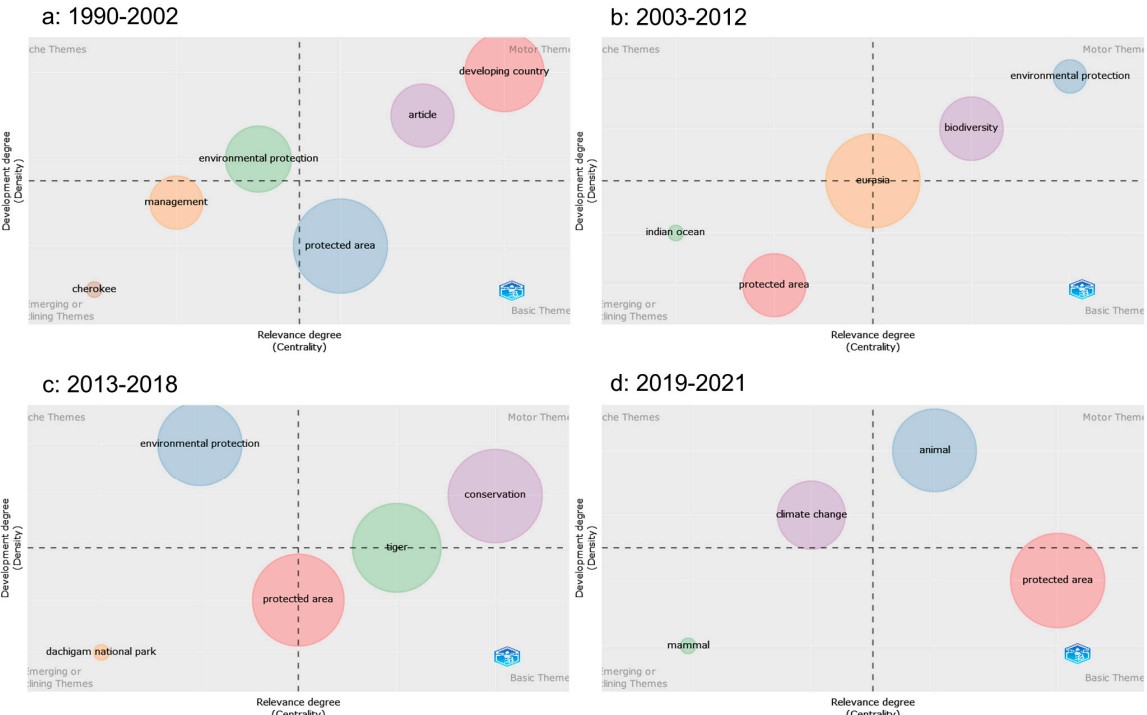

**Figure 9.** Strategic diagrams of the IPAP-related publications in 1990–2002 (**a**), 2003–2012 (**b**), 2013–2018 (**c**), and 2019–2021 (**d**).

"Protected area", which was the basic theme in the first subperiod, remained the basic theme of IPAP in the last two subperiods, indicating the key role of research related to "protected area" in the field. Considering the trends in "tiger" in the third subperiod and "animal" in the fourth subperiod, the research focus was on the topic of "animal" in the last decade. Research related to animals played a crucial role in the field in recent years.

Figure 10a outlines the thematic evolution of the CPAP-related publications. Some of the initial themes were split and recombined into themes for the next subperiod. For example, "nature reserve", which emerged in the second subperiod, was a combination of "Ailuropoda melanoleuca", "conservation", "environmental protection", and the "protected area" in the first subperiod. In addition, a few research topics were integrated into larger research fields. For example, research related to "climate change" and "forestry" were respectively integrated into "endangered species" and "protected area" in the third subperiod. Depending on the evolution of the theme, "environmental protection", "article", "protected area", and "nature reserve" constituted the key elements of the thematic research. In the middle stage, "climate change", "endangered species", and "remote sensing" were hot topics. The emergence of "biodiversity" and "human activity" in the fourth subperiod are new topics in recent years and represent important directions for future research.

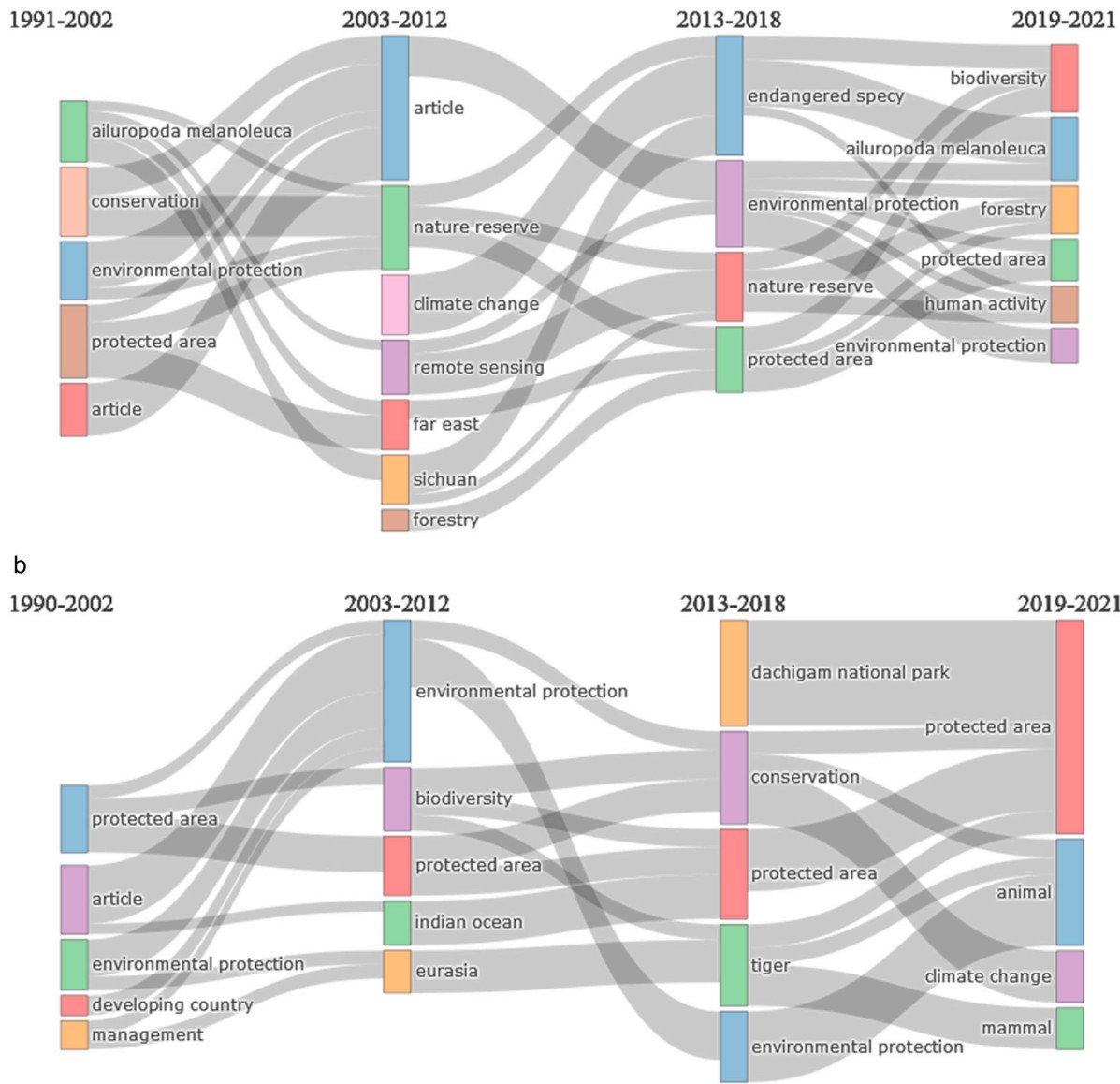

**Figure 10.** Thematic evolution of CPAP-related publications (**a**) and IPAP-related publications (**b**).

According to the thematic evolution of the IPAP-related publications (Figure 10b), "environmental protection" in the second subperiod integrated with "protected area", "article", "developing country", and "management", resulting in comprehensive studies investigating environmental protection. In the second subperiod, "biodiversity" evolved from "protected area", and subsequently developed into various research subthemes, such as "conservation", "protected area", and "tiger". "Animal", "mammal", and "climate change", all of which emerged in the fourth subperiod, the first two from the development and evolution of "tiger" and the third from "conservation". The thematic evolution revealed "environmental protection" and "protected area" as the main topics and basic framework of the thematic research. In both the third and fourth subperiods, "biodiversity", "Indian Ocean", "Eurasia", "Dachigam National Park", and "tiger" were hot topics. The recent emergence of "animal" and "climate change" as new topics will be key for future research.

## 4. Discussion

The concept of PAs has existed since ancient times. The past three decades have seen a pronounced expansion of PAs. The modern PA movement originated in North America, Australia, Europe, and South Africa in the 19th century, mainly focusing on protecting wildlife and stunning natural features. Currently, PAs are expected to meet a variety of ecological, economic, and social goals [36]. The management of PAs is associated with a range of challenges, especially in developing countries [37]. As the two developing countries with the world's largest populations and extraordinary biodiversity [10,20], China and India are both facing intense pressure of balancing social and environmental development [38]. The challenges in PA policy are further compounded by philosophical, geographical, and cultural differences. To address this issue, policy formulation and implementation play a key role. Numerous studies addressing PA policies have materialized due to the shortage of effective political commitments. A comprehensive analysis of these studies can provide a glimpse into the differences in policy focus of the two countries.

The present study compared the major knowledge domains and emerging trends in PA policy research in China and India, highlighting their divergence and directions for future research. Both China and India showed an upward trend in the annual publications related to PA policies. In 2015, after the adoption of the 2030 Agenda [39], the number of CPAP publications surpassed those of IPAP. This implies that, after 2016, CPAP studies entered a vigorous stage, rapidly catching up with IPAP. However, almost every year, the total number of citations of IPAP was higher than that of CPAP. Therefore, CPAP studies are not as recognized as IPAP research. Moreover, Dinghushan National Nature Reserve, the first nature reserve of China, was established in 1956, 20 years later than the first national park in India [6,20]. Studies involving IPAP are relatively more advanced than those of CPAP and also started earlier.

Undoubtedly, national studies accounted for most of the research in PAs. The USA, the UK, and Australia contributed the most to the PAs of the two countries (excluding the home countries themselves). This may be attributed to the key role of these countries in the modern PA movement, thus generating further experience in PAs [40]. The degree of collaboration in IPAP between countries/regions is higher than that of CPAP. This implies that India may have a more inclusive and open attitude toward international collaboration in domestic PA. The number of high-yield authors studying CPAP is higher than those studying IPAP. Furthermore, institutional collaboration in CPAP increased since 2018, while that of IPAP generally occurred prior to 2016. This also implies that IPAP-related research started earlier than CPAP-related studies.

On the basis of the various acts and regulations related to nature reserves enacted and enforced in China and India before 1990, we found that both China and India were concerned with forestry conservation and wildlife protection. However, India enacted these legislations earlier than China [41]. India's establishment of PAs is largely based on the protection of wildlife. The Wildlife Protection Act empowers the State Governments to declare an area as a Sanctuary or National Park [42]. The Indian government attaches

great importance to the protection of wildlife [43], suggesting that wildlife management is one of the most important themes of IPAP investigation. The implementation of forest conservation (Indian Forest Act, IFA) and wildlife protection (Wildlife Protect Act, WA) in India shifted the focus of attention on wildlife management in 1990 and forestry management in 1992 [44]. After 1995, "protected area" and "conservation" in India were high-frequency keywords [45]. The Biological Diversity Act, Wildlife Conservation Strategy, Scheduled Tribes, and Other Traditional Forest Dwellers (Recognition of Forests Rights) Acts were enacted from 1990 to 2010 [46,47]. Accordingly, research related to IPAP focusing on endangered species was extended to management of nature reserves [48]. Studies investigated conservation management extensively, especially the conservation of national parks [49]. The most important issue in the management was the conflict between people and wildlife [50]. Human activities disrupted the migration of wildlife, forcing them to enter human habitats [51]. However, due to the population pressures in India [52], the government's afforestation program led to the confiscation of local farmlands and grazing lands, and the entry of people into the forest areas, leading to a series of wildlife attacks [53]. Compared with CPAP, IPAP studies apparently encompassed more people-related themes, namely, "population density", "land use", and "ecosystem services". The sustainable development of national parks has addressed the livelihood needs of the local population. The Scheduled Tribes and other Traditional Forest Dwellers (Recognition of Forest Rights) Act requires national park authorities to incorporate the interests of local communities into the management via community negotiations [54].

Most of the policies in China were developed later than in India. Since 2000, the Chinese government reformed forestry policies leading to a new phase in the development of PAs [55]. The Natural Forest Protection Program, the Conversion of Cropland to Forest Program, and the Wildlife Conservation and Nature Reserves Development Program (WCNRDP) were formulated to address challenges encountered in forest conservation and restoration [55]. Furthermore, under the concept of "green mountains and clean waters are gold and silver mountains" [56], the Chinese government focused on environmental protection, to usher ecological civilization in China. Studies related to CPAP extended to new research hotspots, such as policymaking and decision making, all of which are linked to strategic policies [57,58]. The dual map also demonstrates the higher number of economic and policy-related articles involving CPAP. Policymaking and implementation management are more important in China than in India. More than 10 types of PAs in China before 2018 included nature reserves, forest parks, wetland parks, scenic spots, marine special reserves, and geological parks, managed by different Ministries and Departments [59]. Laws and regulations were enacted, such as Regulations of Nature Reserves, Management Strategies for Land Inside Nature Reserves, and Management Rules for Marine Nature Reserves. This has led to considerable duplication of administrative work, fragmentation of valuable conservation expertise, and a lack of clarity about rights and responsibilities [58]. To address these issues, the Chinese government has been implementing institutional reforms since March 2018. In June 2019, the General Office of the Central Committee, along with the General Office of the State Council, published a "Guideline to Establish the Mechanism of Natural Protected Areas with National Parks as a Major Component" [60], to re-establish a unified, regulated, and efficient PA management system.

Despite the progress achieved, the current study still had some limitations. Firstly, the study focused on WoSCC and Scopus, thus missing vital information from other databases. Information from other databases will be included in future research. Secondly, most of the results were generated using machine algorithms. Although we made every effort to improve the search strategy, machines are still slightly inadequate compared with artificial induction. Thirdly, the study focused only on published articles, collaborations, and keyword analysis, without examining group policies. Lastly, the comparative analysis of China and India was only based on a literature review and, hence, was not comprehensive enough.

## 5. Conclusions

The annual publications involving the PA policies of China and India show an upward trend. After 2016, the number of publications involving PA policies of China exceeded that of India, while this was not the case for the number of citations. Studies investigating the PA policies of India were started relatively earlier and were more developed than those of China. Key research themes for India included wildlife management and people-related topics, while China focused more on environmental protection and strategic policies. "Biodiversity" and "human activity" are new topics in China and represent important directions for future research. For India, the future research might be around "animal" and "climate change". The protected area system in China is gradually being improved. It is important to reflect on the conflict between wildlife and humans in India. The Chinese government needs to learn from India's experience and safeguard the livelihood of local people while protecting the environment.

**Supplementary Materials:** The following supporting information can be downloaded at https://www.mdpi.com/article/10.3390/cli11010022/s1: Table S1. Annual number and citation of CPAP-related publications from 1990 to 2021; Table S2. Annual number and citation of IPAP-related publications from 1990 to 2021.

**Author Contributions:** Conceptualization, G.W. (Guangyu Wang), W.G. and J.H.; methodology, W.G. and C.Y.; software, C.Y.; formal analysis, Q.Q., C.Y. and A.S.; writing—original draft preparation, W.G., J.H. and S.A.; writing—review and editing, W.G.; visualization, W.G. and J.H.; supervision, G.W. (Guangyu Wang) and G.W. (Guibin Wang). All authors have read and agreed to the published version of the manuscript.

**Funding:** This work was financially supported by the Asia-Pacific Network for Sustainable Forest Management and Rehabilitation (APFNet) (2017SP2-UBC), the China Scholarship Council (CSC) (No. 202108680002, No. 202110230003, No. 202008440171, and No. 202006510054), and three visiting scholarships from Nanjing Forestry University.

**Data Availability Statement:** All datasets presented in this study can be found within the article and in the Supplementary Materials.

**Acknowledgments:** W.G., J.H., Q.Q. and C.Y. are grateful for support from the CSC (China Scholarship Council) Scholarship (No. 202108680002, No. 202110230003, No. 202008440171, and No. 202006510054).

**Conflicts of Interest:** The authors declare no conflict of interest.

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
