# Peer review of "Conservation and Management of Protected Areas in China and India: A Literature Review (1990–2021)"

_climate, doi:10.3390/cli11010022_

Round 1

Reviewer 1 Report

I enjoyed reading this paper; it provides information to fill the gaps between the two countries on conservation and management. However, the future needs more clarification about some areas.

a) What is the main reason behind doing this kind of review rather than direction for the future    

b) How did you fix the time  1 Jan 1990?

c)We need more information or details about the methodology part of the study. For example, I felt this study only used two keywords (e.g., panda reserve + tiger reserve) for searching past studies.  

d) For me very difficult to figure out Fig 2. And my other question is, how come medical, clinical, immunology, and other medical terms to Fig 2?

e)  Conclusion is insufficient for this kind of analysis study. Therefore, for example, better to include future research areas between these two countries.  

Author Response

Thank you so much for your kind suggestions, our reply to your questions is as follows.

Point 1: What is the main reason behind doing this kind of review rather than direction for the future.

Response 1: The development backgrounds of the two countries' protected areas are very different, and there is a need to summarize the past and the current situation for assessment before figuring out the future, and a systematic review can help point the direction of the future.

Point 2: How did you fix the time 1 Jan 1990?

Response 2: Since we wish to summarize the recent decades in this paper, and the amount of relevant literature published before 1990 is almost nil, especially for China, we set the date as January 1, 1990.

Point 3: We need more information or details about the methodology part of the study. For example, I felt this study only used two keywords (e.g., panda reserve + tiger reserve) for searching past studies. 

Response 3: Our search strings are shown in table1, “National Park OR Protected Area OR Nature Reserve OR Tiger Reserve OR Wildlife Sanctuary AND Policy OR Approach OR Strategy OR Program OR Guideline” (for India) and “National Park OR Protected Area OR Nature Reserve OR Panda Reserve OR Wildlife Sanctuary AND Policy OR Approach OR Strategy OR Program OR Guideline” (for China). Panda reserve and tiger reserve are specially added, as they are important parts of PA systems in China and India. The corresponding changes have been made to the text (refer to Line #98 and Table 1).

Point 4: For me very difficult to figure out Fig 2. And my other question is, how come medical, clinical, immunology, and other medical terms to Fig 2?

Response 4: We added more description of Fig.2. The base map on the left is discipline clusters generated by mapping the citing journals, and discipline clusters of the cited journals are on the right. Research publications are therefore listed on the left and the references are listed on the right (refer to Line #181-184).

Medical, clinical, immunology, and other medical terms are discipline clusters of journals and are not in the citation paths, which means PA conservation and management in China and India are not very relevant to these disciplines

Point 5: Conclusion is insufficient for this kind of analysis study. Therefore, for example, better to include future research areas between these two countries.  

Response 5: “Biodiversity” and “human activity” are new topics in China and represent important directions for future research. For India, future research might be around "animal" and "climate change". We added this part to the conclusion (refer to Line #486-488).

Reviewer 2 Report

The peer-reviewed article provides an overview of 1301 and 1263 publications, from India and China, respectively, indexed in Web of Science and Scopus, related to assessing the key role of protected areas in biodiversity conservation. The authors used Biblio-metrix an R package, VOSviewer software, Java application CiteSpace for bibliometric analysis and visualization of information and identified trends. The first thing that the authors of the article found out was a significant increase in publications on this topic in China and a high citation of publications by researchers from India.

The problem studied by the authors is relevant for many countries of the world, as there is a growing concern about the conservation of natural resources and biodiversity around the world.

However, the key words and terms given by the authors refer not only to the problem of the role of protected areas in the conservation of biodiversity, but also to other problems: climate change, the provision of ecosystem services. It may not be entirely correct to confuse all environmental issues in the study of one particular topic.

However, in general, I believe that the researchers obtained interesting and fairly objective results and made a correct analysis of the results.

Author Response

Point 1: the key words and terms given by the authors refer not only to the problem of the role of protected areas in the conservation of biodiversity, but also to other problems: climate change, the provision of ecosystem services. It may not be entirely correct to confuse all environmental issues in the study of one particular topic.

Response 1: Thank you so much for your valuable feedback. In the results, there are such keywords as climate change and ecosystem services. This may be because those environmental issues cover a wide range of keywords, resulting in too many related keywords when clustering. This means that these keywords are related to the research topic to a certain extent, but the detailed content of those environmental issues keywords may need further research.

Reviewer 3 Report

The paper is interesting, and the authors have done good work. However, the paper can be still improved. 

1. In introduction part: Line 53 to 55, data for plant and animal species have been mentioned for India, similar data for China needs to be added.

2. Line 57: Among the top of the world....please mention rank for both the countries if applicable

3. The objective of the study is not clearly spelled out. 

Author Response

We appreciate a lot for your kind recommendations.

Point 1: In introduction part: Line 53 to 55, data for plant and animal species have been mentioned for India, similar data for China needs to be added.

Response 1: Thank you for your suggestion, we added data for plant and animal species in China. China harbors more than 35,000 species of higher plants, 2,700 terrestrial vertebrates, and 28,000 marine species. The corresponding changes have been made to the text (refer to Line #55-56).

Point 2: Line 57: Among the top of the world. Please mention rank for both countries if applicable

Response 2: We are sorry that we found no official data about the accurate ranking, therefore we changed the sentence to “they are both among the most biodiverse countries in the world.” and added the relevant reference (refer to Line #57-58).

Point 3: The objective of the study is not clearly spelled out.

Response 3: This research aims at looking back at the history of China and India's protected area, deepening the understanding of the original causes of their current development status. Our findings could be implied to the protected area policy-making of similar countries and make contributions to global biodiversity conservation. We added this statement of the objective in the introduction (refer to Line #82-85).

Round 2

Reviewer 1 Report

Happy to accept